# Biochemical and Immunological implications of Lutein and Zeaxanthin

**DOI:** 10.3390/ijms222010910

**Published:** 2021-10-09

**Authors:** Javaria Zafar, Amna Aqeel, Fatima Iftikhar Shah, Naureen Ehsan, Umar Farooq Gohar, Marius Alexandru Moga, Dana Festila, Codrut Ciurea, Marius Irimie, Radu Chicea

**Affiliations:** 1Institute of Industrial Biotechnology, Government College University Lahore, Lahore 54000, Pakistan; javariazafar614@gmail.com (J.Z.); amna.aqeel45@gmail.com (A.A.); microbiologisthashmi@gmail.com (F.I.S.); ehsannoreen@gmail.com (N.E.); dr.mufgohar@gcu.edu.pk (U.F.G.); 2Faculty of Medicine, Transilvania University of Brasov, 500036 Brasov, Romania; moga.og@gmail.com (M.A.M.); marius_irimie2002@yahoo.com (M.I.); 3Radiology and Maxilo Facial Surgery Department, Iuliu Hatieganu University of Medicine and Pharmacy, 400012 Cluj Napoca, Romania; 4Faculty of Medicine, “Lucian Blaga” University, 550169 Sibiu, Romania; radu.chicea@gmail.com

**Keywords:** macular carotenoids, antioxidants, bioavailability, genetic engineering, CRISPR/Cas9, lutein binding protein

## Abstract

Throughout history, nature has been acknowledged for being a primordial source of various bioactive molecules in which human macular carotenoids are gaining significant attention. Among 750 natural carotenoids, lutein, zeaxanthin and their oxidative metabolites are selectively accumulated in the macular region of living beings. Due to their vast applications in food, feed, pharmaceutical and nutraceuticals industries, the global market of lutein and zeaxanthin is continuously expanding but chemical synthesis, extraction and purification of these compounds from their natural repertoire e.g., plants, is somewhat costly and technically challenging. In this regard microbial as well as microalgal carotenoids are considered as an attractive alternative to aforementioned challenges. Through the techniques of genetic engineering and gene-editing tools like CRISPR/Cas9, the overproduction of lutein and zeaxanthin in microorganisms can be achieved but the commercial scale applications of such procedures needs to be done. Moreover, these carotenoids are highly unstable and susceptible to thermal and oxidative degradation. Therefore, esterification of these xanthophylls and microencapsulation with appropriate wall materials can increase their shelf-life and enhance their application in food industry. With their potent antioxidant activities, these carotenoids are emerging as molecules of vital importance in chronic degenerative, malignancies and antiviral diseases. Therefore, more research needs to be done to further expand the applications of lutein and zeaxanthin.

## 1. Introduction

Nature is a dexterous and prolific chemist cataloging 100,000 natural products from plants and microbes. This assortment has noticeably influenced the health and well-being of living beings by treating infectious diseases for thousands of years. Their tremendously diverse counterparts have been employed as colorants, fragrances, spices, aphrodisiacs, toxins, and cosmetics. Dietary carotenoids that entail natural lipophilic pigments with an assembly of C40H56 have an extensive presence in nature and are also significant contributors to different realms of life [1]. This astounding structural diversity has made them capable of evolving ≥750 naturally occurring carotenoids [2] which play a significant role in guarding the photosynthetic apparatus from surplus light energy in plants and also participates in light harvesting. While in humans and animals, certain cyclic oxycarotenoids, i.e., lutein (L) and zeaxanthin (Z), are vastly accumulated as macular pigments in the retina and act not only as an imperious precursor for vitamin A, natural colorants, additives in food or feed industry but also as an essential non-toxic chemo-preventive agent against human cancer by modifying immunological responses [3].

Lutein (3R,3′R,6′R-β,ε-caroten-3,3′-diol) and zeaxanthin (3R,3′R-β,β-caroten-3,3′-diol) are considered as the key macular carotenoids that consist of the basic C40 isoprenoid moiety along with ionone ring at each terminal end carrying hydroxyl groups at 3 and 3′ positions. Apart from these two, meso-zeaxanthin has also been a glare of concern as it is an intermediary isomer of lutein but carries a slightly varying structure from lutein and zeaxanthin [4]. These carotenoids in the retina, especially with higher concentration in the foveal groove (0.1–1 mM), act as antioxidants or the blue light filters against oxidative stress, which is a consequence of excessive light exposure [5]. The ratio of lutein: zeaxanthin: meso-zeaxanthin in the peripheral retina measured by HPLC is about 3:1:0. However, the concentration of aforementioned carotenoids rises 100-fold in macular lutea and the ratio changes to 1:1:1 [6].

Apart from retina, lutein is present in a number of human tissues. They make up about 0.1–1.23 μM level of serum, 0.1–3.0 μM in the liver, 0.037–2.1 μM in kidney and 0.1–2.3 μM in lungs [7]. However, lutein and zeaxanthin cannot be fabricated de novo, even though *Homo sapiens* devour ~50 carotenoids in the diet, and ≤20 different carotenoids are usually found in the serum [8]. Usually, green leafy vegetables abundant in lutein and zeaxanthin maintain their level in the retinal tissues, serum, and adipose tissues [9]. The concentration of carotenoids in serum tends to reflect their most recent intakes, whereas adipose tissues are the good indicator of long-term inputs. Since, these macular carotenoids have a decisive role in preventing ophthalmological disorders like age-related macular degeneration (AMD), retinitis pigmentosa, and cataracts [10]. The world concern has shifted its concern towards the formulation of L and Z rich products to exploit their role in therapeutic applications.

The global market for carotenoids has previously been expected to grow at approximately US$1400 million in the year 2017. It would reach $2000 million by 2022 with an annual growth rate of 5.7% during 2017–2022 [11]. In this regard, lutein extricated from Marigold Flowers (*Tagetes erecta*) has acquired a top status in the fastest-growing carotenoid market. But the lutein content of Marigold flowers is near to the ground (0.3 mg^−1^ DW), and hence microbes are being looked up for an alternative source of this xanthophyll [12]. The global market expects a reach of EUR 409 million by 2027 for lutein at a Compound Annual Growth Rate (CAGR) of 6.10% over the projected time frame of 2020–2027 [13]. Likewise, the global zeaxanthin value is expected to reach ~US$ 210 Million by 2030, at a CAGR of 8.2% [14] (Figure 1). However, the lutein and zeaxanthin application in the food industry is limited because of the unstable nature and chemical changes usually faced during food processing [15].

Various techniques are being employed to minimize the loss of carotenoid content during food processing and storage. These mainly include freezing, addition of antioxidants and removal of oxygen in a vacuum-sealed/airtight container [16]. In addition, protein engineering techniques have recently been employed to enhance the shelf-life and activity of lutein and zeaxanthin. This review will highlight the bioavailability of lutein and zeaxanthin in different domains of life, along with the engineering of these carotenoids to make them applicable for healthcare and industrial applications.

## 2. Lutein and Zeaxanthin in Different Kingdoms of Life

The carotenoids derived from plastids in plants are generally isoprenoid-derived molecules. Here, several nuclear-encoded enzymes aid the production of these carotenoids [17]. Usually, the biosynthesis of carotenoids takes place through two different mechanisms. These involves the well-explained mevalonate pathway (MVP) in florae [18], and recently studied methylerythritol 4-phosphate (MEP) pathway that often take place in eubacteria including *Escherichia coli* (shown in Figure 2) [19]. An alternative or non-mevalonate pathway, known as the deoxyxylulose-5-phosphate (DOXP) pathway or MEP pathway, tends to produce isoprenoid precursors, i.e., isopentenyl diphosphate (IPP) and dimethylallyl diphosphate (DMAPP) in plant chloroplast, algae and cyanobacteria etc. that ultimately reach the carotenoid synthesis pathway [20]. In addition, Arigoni et al. [21] have also illustrated the non-mevalonate pathway in *Catharanthus roseus* for the biosynthesis of lutein, β-carotene, and phytol.

### 2.1. Biosynthesis and Presence of Lutein and Zeaxanthin in Kingdom Plantae and Animalia

Photosystem II is a critical player in photosynthesis, and its photo-protection is thus a matter of utmost importance to alleviate the light-induced damage mediated by the generation of reactive oxygen species (=photo-oxidative stress). Naturally occurring carotenoids are notable for their ability to neutralize the effect of triplet chlorophyll (3 Chl*) and singlet oxygen (1 O_2_*) [22]. Usually, Carotenoids having cyclic end groups and β- and ε-rings are consistently prevalent in the reaction centers of photosynthetic organisms like algae, plants, and cyanobacteria. The catalytic machinery accountable for catalyzing the generation of β- and ε-rings is encoded by distantly related (36% identity for the deduced amino acid sequences) single-copy genes of *Arabidopsis thaliana* [23]. Thereof, carotenoids with β- and one ε-ring are regarded as lutein in floral species. In this regard, Bialek-Bylka et al. [24] have stated the prevalence of lutein’s central cis-isomer in a prominent light-harvesting domain of Photosystem II of higher plants (e.g., spinach). Here, these carotenoids have played a significant role in quenching triplet chlorophyll and thus playing a substantial role in photo-protection.

The lutein production from lycopene follows four staged enzymatic reactions. These reactions involve β- and ε-ring cyclization and hydroxylation of each ring at the C-3 position. For this purpose, three enzymes have already been identified in *Arabidopsis* and regarded as carotenoid hydroxylases. The two non-heme di-iron β-ring monooxygenases (the B1 and B2 loci) are involved in catalyzing the hydroxylation of the β-ring in β,β-carotenoids, and one heme-containing monooxygenase (CYP97C1, the *LUT*1 locus) catalyzes the hydroxylation of ε-ring of β, ε-carotenoids [25]. However, recently, it has been documented that *Arabidopsis* CYP97A3 (the *LUT*5 locus) encodes another carotenoid hydroxylase with distinct in vivo activity towards the β-ring of α-carotene (β,ε-carotene) and negligible activity on β-rings of β-carotene (β,β-carotene). Hence, Kim et al. [26] have reported a most anticipated pathway for lutein synthesis in plants (as shown in Figure 3): that follows the ring cyclizations for α-carotene production. After the hydroxylation of β-ring at α-carotene by CYP97A3 for the generation of zeinoxanthin, succeeded by ε-ring hydroxylation of zeinoxanthin by CYP97C1, lutein can be harvested.

Furthermore, the carotenoids having two β-rings, i.e., β-carotene and zeaxanthin, are also omnipresent and are contemplated as the photo-protective agents against the excess light. There is a similarity in the mainstream mechanism of lutein synthesis, ring cyclization followed by hydroxylation, to its neighboring structural isomer, zeaxanthin. Usually, non-heme B1 and B2 β-ring hydroxylases are involved in the formation of zeaxanthin, while lutein results from P450-mediated α-carotene ring hydroxylases activity [26]. With a momentous role in protecting chlorophyll, zeaxanthin is also a key player in the lipid bilayer of the plant membrane (or most probably at the protein-membrane interface) [27].

However, the bioavailability of lutein and zeaxanthin may differ on the basis of vegetable or fruit type (Table 1). The lutein concentration in the natural sources relies on the kind, maturity level, processing technique, and storage time of food [28]. Interestingly, only traces of zeaxanthin have been found in most dark green vegetables, including scallions, green lettuce, celery, spinach, and brussels sprouts [29]. However, the qualitative portrayal of lutein in green vegetables is essentially the same except for romaine lettuce that carries a significant amount of another dihydroxycarotenoid, lactucaxanthin (ε,ε-carotene-3,3′-diol) [30].

Unlike plants, animals cannot synthesize carotenoids but can receive them from diet and transform them into various forms. The oxidative metabolites of lutein; papirioerythrinon, philosamiaxanthin (3-hydroxy-β,ε-caroten-3′-one), and fritschiellaxanthin exist in various species of butterflies and moths [31]. Maoka et al. [32] investigated *Heteroptera* (assassin bugs, stink bugs, water scorpions, water striders, backswimmers, and water bugs) from a chemo- systematic and chemical ecological perspective. It appeared that lutein, β-carotene and, β-cryptoxanthin procured from plants were prevalent in both stink bugs and leafhoppers.

Similarly, Czeczuga [33] has reported a number of carotenoids in multiple species of *Lepidoptera.* These include lycopene, zeaxanthin, β- cryptoxanthin, and lutein epoxide. The most prevalent carotenoid in butterflies, known as zeaxanthin, is formed from the β-carotene or β-cryptoxanthin, absorbed from food, as its epoxide form is less dominant in higher plants than lutein. Furthermore, dragonflies of the order *Odonata* have gained attention for their carotenoid content and courage, strength, victory and happiness in Japan [34]. In their larval form, the dragonflies, i.e., *Anisoptera* and *Zygoptera,* carry lutein, β-carotene, β-cryptoxanthin, and fucoxanthin. Dragonfly larvae nourishing on aquatic creatures; tadpoles, water fleas, and small fish, derive their carotenoid content from them. The adult dragonfly of *Anisoptera* and *Zygoptera* carry a significant portion of β-carotene along with lutein, zeaxanthin, echinenone, and β-caroten-2-ol [35]. Hence, food chain investigations suggest that adult dragonflies carry lutein and zeaxanthin from their feed which includes flies, butterflies, mosquitoes, moths, spiders, and plant-hoppers.

**Table 1 ijms-22-10910-t001:** Occurrence of lutein and zeaxanthin in different parts of plants.

Source	Common Names	Concentration of Lutein (mg/100g)	Concentration of Zeaxanthin (mg/100g)	Cumulative Concentration of Lutein + Zeaxanthin (mg/100g)	General Applications	Reference
**Vegetables:**						
*Asparagus officinalis* L.	Garden asparagus/Sparrow grass	Cooked asparagus		0.382–0.653 (in raw asparagus)	Antioxidant property	[36,37]
		0.991	0.22			
*Asparagus acutifolius*	Wild asparagus	0.544–1.913	N.A	N.A	Nutritional supplements	[38]
*Brassica oleracea var. italica*	Broccoli	Cooked broccoli		1.079	Antioxidant property	[39,40]
		3.11–3.960	1.73628			
*Solanum nigrum* L.	European black nightshade	84.38	N.A	N.A	Nutritional supplements	[41]
*Hibiscus cannabinus* L.	Kenaf	104.24	4.59	N.A	Antioxidant property	[42]
*Capsicum annuum*	Green chilies	1.902	0.06285	13.74	Antioxidant property	[43]
*Solanum tuberosum*	Potato	0.1352–0.1524	0.0077	N.A	Nutritional supplement	[44,45]
*Daucus carota* L.	Wild carrot	40.17	0.59	N.A	Antioxidant property	[37]
*Lactuca sativa*	Lettuce	3.824	N.A	2.313	Ophthalmological significance	[37]
**Herbaceous parts**						
*Ocimum basilicum* L.	Basil	4.2–8.3	0.2–0.6	6.634 ± 0.410	Nutritional supplements	[46,47]
*Petroselinum crispum*	Parsley	4.326	1.236	5.562	Nutritional supplements	[48]
*Coriandrum sativum*	Coriander	9.920	N.A	4.740 ± 4.36	Adjuvants in food preparation + antioxidants	[43,49]
*Mentha spicata*	Spearmint	17.74	0.28	18.00	Nutritional supplement	[41,43]
*Elattaria cardamomum*	Green or true cardamom	0.44	N.A	0.35	Antioxidant property	[50]
**Fruits**						
*Malpighia emarginata*	Barbados Cherry	9.20 ± 0.23	1.14 ± 0.03	N.A	Antioxidant property	[51]
*Cucurbita maxima*	Pumpkin	10.62	0.278	N.A	Antioxidant property	[50]
*Solanum lycopersicum*	Tomato	0.289	0.0144	94	Antioxidant property	[37]
*Citrus X sinensis*	Orange	0.033	0.029	0.129	Antioxidant property	[37]
*Vitis vinifera ‘Red Globe’*	Red grapes	0.024	0.004	N.A	Antioxidant property	
**Seed/Grain**						
*Zea mays* L.	Corn	1.47	0.01031	0.01662–0.02057	Antioxidant property	[52,53]
*Triticum durum*	Whear	1.5–4.0	0.00049	N.A	Nutritional supplmenets	[54,55]

### 2.2. Prevalence of Lutein and Zeaxanthin in Kingdom Protista- Algae and Fungi

Microbes are notable for assembling various bioactive compounds that possess an exceptional therapeutic potential and stand out as an emerging source of profitable secondary metabolites. The carotenoids present in microalgae have their role as accessory pigments in the photosystems and act as the fundamental unit of light-harvesting systems with a role in phototaxis and as photo-protection [56]. Lutein is present within the members of *Chlorophyta*, *Cryptophyta*, *Chlorarachniophyta*, *Rhodophyta,* and *Euglenophyta* [57]. Del-Campo et al. [56] have reported the prominent lutein content in fifteen strains of *Chlorophyceae*, including *Chlorella fusca* SAG 211-8b, *Chlorococcum citriforme* SAG 62.80, *Muriellopsis* sp., *Neospongiococcum gelatinosum* and *Chlorella zofingiensis* CCAP 211/14. Moreover, microalgae, such as *Chlorella zofingensis* and *Chlorella protothecoides* with a lutein concentration of 0.13%, and 4.6% per dry weight, have emerged as the alternative reservoirs of lutein [58]. Nevertheless, members of the class *Bangiophyceae* in red algae, such as the *Porphyra/Pyropia* species, have a carotenoid profile dominated by lutein along with α-carotene, β-carotene, and zeaxanthin [59]. The red algae are characterized for utilizing the DOXP pathway to synthesize of lutein and zeaxanthin (as shown in Figure 3). Similarly, the order *Euglenida* has been under the spotlight for its potential as a feedstock to produce biodiesel and carotenoids. Recently, Deli et al. [60] have reported the production of 23.7% (*w*/*v*) lutein in *E. sanguinea*, a ubiquitous resident of various shallow and eutrophic freshwater bodies. In addition, certain growth conditions (e.g., temperature, pH, irradiance, availability, nitrogen source, salinity or ionic strength) also favor the high carotenoids production (up to 13% *w*/*w*) in the green algae *Dunaliella salina.* This alga assembles β-carotene, zeaxanthin, astaxanthin, cryptoxanthin and lutein [61,62]. Table 2 demonstrates the microalgal species that are involved in the fabrication of lutein and zeaxanthin.

Fungi is another alternative source of natural pigments due to their multiple advantages over plants. These include fast and easy growth, season-independent production of stable and soluble pigments, and varying shades of color [77]. Numerous filamentous fungus generates various carotenoids and xanthophylls (e.g., lycopene, carotene) (e.g., astaxanthin, lutein, zeaxanthin, and violax). Kingdom fungi produce carotenoids via mevalonate pathway, which utilizes 5-carbon isopentenyl pyrophosphate (IPP) as a precursor. This IPP is produced from hydroxymethylglutaryl coenzyme A (HMG-CoA) or derivatives of 1-deoxy-D-xylulose 5-phosphate (DXP) or 2-C-methyl-D-erythritol (MEP) [78].

Over the past two decades, several studies have been carried out on various oleaginous fungi that serve on naturally occurring sources of carotenoids. These include *Neurospora crassa*, *M. hiemalis*, *Blakeslea trispora*, *Phycomyces blakesleeanus,* and *M. circinelloides* [79]. In this regard, Mohamed et al. [80] have performed the comparative investigation of oleaginous *Mucoromycota* Fungi for their carotenoid potential. Genus *Mucoromycota* entails typical oleaginous filamentous fungus that forms valuable carotenoids and essential fatty acids. The investigation revealed that *M. hiemalis* AUMC 6031 and *M. hiemalis* AUMC 6695 took lead by the highest yield of total carotenoids (640 μg/g) with a similar profile of zeaxanthin (34%), *β*-carotene (65%), canthaxanthin, and astaxanthin (5%), in regards of total carotenoids concentration. Similarly, Rodríguez-Sáiz et al. [81] have reported the presence of the *crtS* gene for the formation of zeaxanthin and β-cryptoxanthin in *Mucor circinelloides*. Thus, fatty acids and carotenoids procured from oleaginous microbes have a potential to bring revolution in various therapeutic applications.

### 2.3. Biosynthesis of Lutein and Zeaxanthin in Domain Eubacteria

A vast repertoire of microorganisms is considered as the distinct reservoir of various valuable products. These organisms appear to be a reliable alternative against plant-based carotenoids due to enhanced productivity, availability throughout the year, and ease in maintaining desired conditions during the fermentation process. In addition, transforming a low-cost substrate (industrial or agricultural waste) into high-value carotenoids recapitulate our commitment to turning waste into gold. The prokaryotic nature of bacteria allows genetic manipulations more easily in contrast with the eukaryotic plants and algae. Several promising bacterial strains in freshwater and marine waters have been reported for having myriad of carotenoids. Around 30 bacterial isolates belonging to *Chryseobacterium, Arthrobacter, Zobellia,* and *Flavobacterium* genera identified from King George island, Antarctica, carry ten different carotenoids with the most frequent ones include β-carotene, zeaxanthin, and β-cryptoxanthin [82]. The biological route followed for zeaxanthin extraction is from marigold (*Tagetes erecta*) flowers. Nevertheless, the generated zeaxanthin is of low purity, since the extract contained only 20% of trans-zeaxanthin and varying concentrations of other carotenoids, waxes, fats, and oils [83]. A comparative study has revealed that zeaxanthin isolated from microbial (e.g., *Flavobacterium multivorum*) source has 2–3 times more bio-availability than the less stable ones from marigold flowers [84].

Among these various taxa, the family of *Enterobacteriaceae* (e.g., *Enterobacter* and *Erwinia/Pantoea* genera) are the foremost producers of zeaxanthin and its derivatives such as zeaxanthin β-glycoside [85]. Similarly, Entophytic microbes are a plenteous source of numerous bioactive natural products. Fidan and Zhan [86] have reported an endophytic bacterium, *Pseudomonas* sp. 102515, from the leaves of *Taxus chinensis* which is a significant producer of zeaxanthin diglucoside. Although some of the bacterial species (listed in Table 3) have been identified as zeaxanthin producers, their commercial scale production is yet to be established.

Despite being the potential source of pigments, cyanobacteria are also involved in the large-scale production of phycocyanin from *Arthrospira platensis*. The reason involves the insignificant level of carotenoid content in cyanobacteria compared to vascular plants or microalgae [97]. However, recently Morone et al. [98] have reported the potential of ethanol extracts of picocyanobacterial strains of the genera *Cyanobium* and *Synechocystis* and filamentous strains of the genera *Nodosilinea*, *Phormidium,* and *Tychonema* for skin applications. The obtained carotenoids involve canthaxanthin, β-carotene and echinenone, with zeaxanthin and lutein being the most abundant ones (49.82 and 79.08 µg g^−1^, respectively). Thus, the ethanol extracts obtained from *Cyanobium* and *Tychonema* sp. are identified as the potential candidates for anti-aging formulations, as they trigger the proliferation of fibroblast and inhibit the digestion of hyaluronic acid.

## 3. Enhancing the Production of Lutein and Zeaxanthin by Genetic Engineering Approach

The importance of carotenoids in human health and nutrition has led to various efforts in increasing their level in food crops. Traditionally, lutein is produced through the solvent extraction procedure of lutein esters, mediated with saponification [99]. Khachik [100] has revealed a procedure of purgation and recrystallization of lutein from saponified marigold oleoresin. But this strategy involved several solvents for extraction and purification steps. Another investigation involved the saponification of crude oleoresin from marigold extract using ethanol, water, and 45% alkali for 3–5 h at 45–80 °C [101]. This process had a higher yield but appeared uneconomical due to the high concentration of alkali and the minor content of lutein ester in substrate. Even though, organic solvents have previously been utilized to process biomaterials, there are concerns over their implications in the environment and food industry. Since the transcriptional regulation of carotenogenic genes is a vital strategy for accumulating carotenoids in plants, global attention has been directed towards metabolic/genetic engineering techniques for the enhanced production of lutein and zeaxanthin.

### 3.1. Engineering Biosynthetic Pathway of Lutein by Random Mutagenesis Approach

Altering the growth metabolism of microalgae through genetic engineering techniques is a vital strategy for enhanced lutein production under phototrophic, heterotrophic, and mixotrophic growth conditions [102]. Chen et al. [103] have revealed the random mutagenesis mediated variants of wild type *Chlorella sorokiniana* MB-1 for lutein extraction. By adding 6.0 g/L of sodium acetate under mixotrophic conditions, the *C. sorokiniana* MB-1 wild type strain produced lutein content of 2.39 mg/L per day and 5.86 mg/g per day. This indicate the fact that under same culture condition, the MB-1-M12 mutant strain generated a greater lutein content (7.52 mg/g and 3.63 mg/L/day) as opposed to wild type strain. Thus, developing the microalgal strains with an inbuilt capacity to generate a higher lutein content is crucial for commercialization of microalgal-mediated lutein.

In addition, Cordero et al. [12] have reported that the random mutagenesis by N-methyl-N′-nitro-nitrosoguanidine (MNNG) of *C. sorokiniana* give rise to higher lutein levels. The produced mutants were screened for resistance against the carotenogenic pathway inhibitors, i.e., nicotine and norflurazon. Consequently, mutant MR-16 depicted a higher level of cellular lutein content than the wild-type strains. The produced MR-16 mutant showed nicotine resistance activity by acting as a specific inhibitor of the lycopene β-cyclase that could have a capacity to alter the specific activity of this mutant by adding an altered assembly for the herbicide-binding site. This ultimately resulted in higher enzyme activity and improved production of lutein under the photoautotrophic conditions. Additionally, this mutant’s higher growth rate is attributed to the increased lutein content under photoautotrophic growth conditions than the wild type strain. This carotenoid is notable for its role in photosynthetic assembly’s structure, function and photo-protection. Thus, paving the way towards an efficient photosynthetic pathway.

### 3.2. Engineering Biosynthetic Pathway of Lutein and Zeaxanthin by Gene Assembly Approach

A number of unique carotenoids have been synthesized by assembling carotenoid biosynthetic genes from various living entities and expressing these “hybrid” mechanisms in a recombinant host is a unique approach for tomorrow’s sustainability [104]. By introducing appropriate gene clusters into noncarotenogenic host, *Escherichia coli*, carotenoid production can be improved [105]. Recently, Nishizaki et al. [106] have attempted to optimize the zeaxanthin production by reordering the five biosynthetic genes in *Pantoea ananatis*. In this regard, recently outlined operons for zeaxanthin production were fabricated by the technique of ordered gene assembly in *Bacillus subtilis* (OGAB), which can accumulate multiple genes in one step using an intrinsic *B. subtilis* plasmid transformation system. Increased zeaxanthin formation in *E. coli* (820 g/g [dry weight]) was found in transformants carrying a plasmid with genes arranged in an order of zeaxanthin metabolic pathway (crtE-crtB-crtI-crtY-crtZ). These results strengthen the utilization of the OGAB method as a unique approach to reorder *crt* genes for the production of carotenoids. This pathway opens the door for the application of various operons having desired gene order in investigating metabolic pathways.

Similarly, Choi et al. [85] have also demonstrated that the recombinant *E. coli* with six carotenoid biosynthesis genes (crtE, crtB, crtI, crtY, crtZ, and crtX) of *Pantoea ananatis* can produce a novel carotenoid glycoside that has a characteristic carbohydrate moiety, quinovose (chinovose; 6-deoxy-D-glucose). Later, this carotenoid glycoside was recognized as 3-β-glucosyl-3′-β-quinovosyl zeaxanthin through spectroscopic and chromatographic analysis. These zeaxanthin glycosides were appeared as 3R,3′R aglycones with hydroxyl groups added by the CrtZ enzyme. Later, it was revealed that zeaxanthin synthesized from β-carotene via CrtR or CYP175A1 had the same stereochemistry as CrtZ.

In an attempt to obtain a high yield of zeaxanthin from photoautotrophic prokaryotes like *E. coli*, another system, i.e., *Synechocystis* sp. strain PCC 6803, has come into the limelight that overexpress the biosynthetic genes of carotenoids. This system comprises of the *psb*AII gene, which encodes the highly expressed photosystem II D1 protein with a strong promoter. Upon further examination of the *Synechocystis* genome, it is discovered that three genes, namely *psb*AI, *psb*AII, and *psb*AIII, encode the D1 protein. Even in the truancy of the other two *psb*A genes, *psb*AII and *psb*AIII may express independently and sustain photoautotrophic growth [107]. Thus, the *psbAII* locus can be utilized as an integration system to overexpress the genes residing in *Synechocystis*. Later in the *psb*AII coding sequence, Lagarde et al. [108] introduced the yeast isopentenyl diphosphate isomerase (ipi), β-carotene hydroxylase (crtR), phytoene desaturase and phytoene synthase genes of Synechocystis (crtP andcrtB). Overexpression of the crtP and crtB genes resulted in a 50% increase in the myxoxanthophyll and zeaxanthin content of the mutant strain. Moreover, the overexpression of *crtR* triggered a 2.5-fold increase in zeaxanthin in the mutant strain compared to the wild-type strain. Thus, altering genes expression by inducing over-expression or the deletion of specific genes is a successful approach for changing the carotenoid content of cyanobacteria. Additionally, the assembly method used in this system is capable of replacing genes without introducing antibiotic resistance cassettes. Therefore, the absence of such cassettes in these strains marks the future of the ultimate desire of the biotechnology industry to avoid the spread of antibiotic resistance genes cassettes.

Cyanobacteria produce a vast range of products that mainly include β-carotene, zeaxanthin, echinenone and myxoxanthophyll [109]. In contrast to zeaxanthin, lutein and the α-branch of the MEP pathway are absent in cyanobacteria However, Lehmann et al. [110] created a genetically modified cyanobacterium *Synechocystis* that lacks the MEP α—branch required for lutein synthesis. In this regard, a cassette encompassing four *Arabidopsis thaliana* genes encoding two lycopene cyclases (*AtLCYe* and *AtLCYb*) and hydroxylases (*AtCYP97A* and *AtCYP97C*) was established into a *Synechocystis* strain, that had a dearth of endogenous cyanobacterial lycopene cyclase *cruA*. This specie produced increased amount of lutein, which aided in the investigation of lutein-associated activities such as the assembly of plant-type light-harvesting complexes and other carotenoids-binding proteins.

### 3.3. Enhanced Production of Zeaxanthin by CRISPR/Cas 9 Based Approaches

Zeaxanthin is a remarkable pigment for retina and antioxidant activities. Since zeaxanthin aids in the preclusion of age-related macular degeneration, its utilization at commercial scale for implication in nutrition and therapeutics has been explored most seriously in these times. However, in order to cater with healthcare problems, it is vital to produce zeaxanthin of high purity.

Genetic engineering techniques have wide-ranged applications in the phenomenon of carotenogenesis, as they carry a significant potential to increase the commercial status of carotenoids. Recently, the green alga *Chlamydomonas reinhardtii* has acquired utmost consideration for being a photosynthetic factory, but its role in carotenoid production has not yet been revealed. Over the course of time, the diversity of carotenoid mutants of *C. reinhardtii* has initiated the discovery of genes involved in lutein and zeaxanthin biosynthesis [111].

Since there are several strains of *C. reinhardtii* with exceptional traits but CC-4349 is an incredible host for the pigment and carotenoid formulation. In this regard, Baek et al. [112] have produced the knock-out mutant of the zeaxanthin epoxidase gene with the help of DNA-free CRISPR-Cas9 ribonucleoproteins in CC-4349. Resultantly, the produced mutant depicted a substantially elevated zeaxanthin concentration (56-fold) with increased productivity by 47-folds compared to the wild type without undermining lutein concentration. These findings clearly indicate to the prospect of commercializing micro-algal mutants through DNA-free CRISPR-Cas9 ribonucleoproteins to produce high-value goods in a low cost.

Similarly, Song et al. [113] have revealed a CRISPR-Cas9 ribonucleoprotein mediated double knockout mutant of *C. reinhardtii*. The modification was made to the lycopene epsilon cyclase (LCYE) gene in order to remove the α-branch of xanthophyll biosynthesis in a zeaxanthin epoxidase knockout variant (ZEP). After three days of cultivation, the double knockout mutant produced 60% more zeaxanthin (5.24 mg L^−1^) with a content of (7.28 mg g^−1^) than the acquired parental line. In nutshell, this research entails solutions to overcome the pitfalls of down-stream processes in the production of highly pure zeaxanthin.

### 3.4. Regulation of Biosynthetic Pathways of Lutein and Zeaxanthin by Carotenoid Desaturases Cyclases

In the carotenoid biosynthetic pathway, ζ-carotene desaturase (ZDS) and carotenoid isomerase (CRTISO) allow the conversion of ζ-carotene to the lycopene, which is a substrate for two competing lycopene cyclases, ε-LCY, and β-LCY. In the past, ZDS gene has not been studied to improve the carotenoids content in plants. Additionally, its function in toleratings abiotic stresses has not yet been revealed. Nevertheless, Li et al. [114] have stated that the overexpression of *IbZDS* gene in the *Ipomoea batatas* brings forth a significant increase in the lutein contents (1.74–2.37-folds) and β-carotene (2.24–3.96-folds) contents along with increased salt tolerance in the transgenic sweet potato. Similarly, Phytoene desaturase (PDS), the structural coherent of ZDS, is the key player of lutein biosynthetic pathway. In the biosynthetic pathway of lutein, this enzyme allows the transformation of colorless phytoene to the colorful carotenoids in algae and plants.

In this regard, Li et al. [3] have revealed that by employing random and site-directed mutagenesis in PDS, a single amino acid mutation (N144D) functions in the activity of enzyme. This mutant produced an inactive enzyme, thus implying the fact that amino acid 144 is vital for PDS enzymatic activity. Moreover, Fantini et al. [115] have recently revealed that the virus-induced gene silencing (VIGS) of PDS in tomatoes results in the reduction of 55% total carotenoid content (including lutein), with phytoene and phytofluene as the most prevalent ones.

Moreover, *ε-LCY* and *β-LCY* direct the lycopene flux towards α- and β-branch of the carotenogenic pathway and produce α-carotene and β-carotene [116]. These two cyclases are interlinked closely and stem out through the phenomenon of ancient gene duplication. Several studies of these two cyclases revealed that separate genes in plants encode them. Usually, a single gene encodes a fusion protein of LCYB, LCYE, and a C-terminal light-harvesting complex (LHC) domain in the prasinophyte algae of the order Mamiellale. However, the ability to tune the product ratio of the Mamiellales’ lycopene cyclase fusion protein renders it advantageous for the biotechnological formation of asymmetric carotenoids like α-carotene or lutein [117]. Zeng et al. [118] utilized a post-transcriptional gene silencing technique to down-regulate the expression of *TaLCYB* in transgenic wheat to explore the function of Lycopene β-cyclase in the production of lutein. The results demonstrated a decrease in β-carotene and lutein content that was accompanied by lycopene accumulation to slightly reimburse for the total carotenoid content. This demonstrates the fact that lycopene β-cyclase, carry a pronounced role in lutein biosynthesis.

In addition, the overexpression of lycopene β-cyclase (*IbLCYB2*) in sweet potato, *Ipomoea batatas* (L.) Lam. categorically enhance α-carotene, β-carotene, zeaxanthin, and β-cryptoxanthin content with increased tolerance towards drought, salt, and oxidative stress [119]. Figure 4 illustrates the proposed model for increasing the carotenoids level by overexpressing the LCYB gene in the sweet potato. The Co-overexpression of *dsx* and *idi* genes in the transgenic *E. coli* has a supplementary effect on the zeaxanthin production that reached 1.6 mg/g DCW [120]. Moreover, Li et al. [121] also engineered *Escherichia coli* by utilizing fusion protein-mediated substrate channeling and incorporated tunable intergenic regions to optimize zeaxanthin biosynthesis from lycopene. This approach is more beneficial than protein fusion approach for co-expression of lycopene β-cyclase gene crtY and β-carotene 3-hydroxylase gene crtZ. In a nutshell, the impact of substrate channeling advocates the CrtZ catalyzed reaction to be the rate-limiting step in the biosynthesis of zeaxanthin.

The lycopene ε-cyclase (LCYE), similar to lycopene β-cyclase (LCYB), is also involved in the biosynthetic pathway of carotenoids. Tokunaga et al. [122] also overexpressed *the LCYE* gene in *C. reinhardtii* to determine its aftermaths on lycopene metabolism and lutein synthesis. *Chlamydomonas* transformants also produce an increased amount of lutein per culture (up to 2.6-fold) without decreasing cell yields. The overexpression of *LCYE* increases the transformation of lycopene to α-carotene, which results in lutein production. For alterations in carotenoid profile by RNAi mediated decreased expression of lycopene epsilon-cyclase, (ε-CYC),

Yu et al. [123] reported a shift in carotenoid accumulation within the seeds of *B. napus*. The bio-engineered seeds expressing this construct experienced an elevated level of zeaxanthin, beta-carotene, violaxanthin, and surprisingly lutein. An elevated level of carotenoid content evolved from decreased expression of ε-CYC in seeds implies that this gene is a rate-limiting step in the biosynthetic pathway of carotenoid. Three possibilities comprehensively explain lutein production in an ε-CYC mediated silenced lines of *B. napus*: (1) The presence of additional copies of ε-CYC gene that are not silenced by RNAi may compensate for the formation of ε-ring. The southern blot analysis employing enzymes bearing either single or no recognition sites within the cDNA sequence revealed that the genome of *B. napus* carries at least two homologs of an ε-CYC gene [124]. The identification of these homologs must be done to determine their activity in RNAi silenced and parental tissues. (2) The generation of the ε-ring with the help of enzymes carrying a broad spectrum of the substrate; enzymes bearing the capability of ε-ring formation at the terminals of aliphatic compounds may act as the driving factor for ε-rings formation in lycopene. (3) ε-CYC is not regarded as the rate-limiting gene for lutein biosynthesis in seeds. Still, silencing an ε-CYC gene may result changes in the cellular compartments or sequestration molecules, thus enhancing seed capacity to stockpile lutein.

## 4. Esterification of Lutein and Zeaxanthin to Enhance the Stability and Bioavailability in Harsh Environmental Conditions

Lutein esters are widely recognized as the primary carotenoids in marigold petals, extracts, oleoresins, and derivative products. Indeed, several lutein monoesters and diesters acylated with lauric (12:0), myristic (14:0), palmitic (16:0), and stearic (18:0) acid moieties have been characterized, with the major compounds being lutein dimyristate, dipalmitate, myristic-palmitate, and palmitate-stearate [125]. But for most wheat flour products, color is a significant quality criterion. The main component of a creamy bread wheat flour and various other components is the asymmetric dihydroxy carotenoid, lutein, that contributes mainly to the yellow color of yellow alkaline yellow noodles (YAN) [126]. Lutein’s role in the yellow and creamy color of end products and contribution towards health and nutrition [127,128] is well documented. However, lutein is degraded directly by heat and light and indirectly by free radicals produced from lipid oxidation [129,130].

Identifying the gene(s) essential for xanthophyll esterification would be beneficial for breeding purposes, as esterified carotenoids have a greater capacity for accumulation inside plant cells and are more stable in the course of post-harvest stowage. In this context, Requena-Ramrez et al. [131] examined five genes in *H. chilense* genome that were identified as potential candidates of xanthophyll acyltransferase (XAT) in lutein esterification. All of these genes expressed themselves in tritordeum during grain growth, but only XAT-7Hch was highly upregulated. Notably, its wheat orthologue TRAESCS7D02g094000 (XAT-7D) has recently identified as the gene encoding XAT enzyme for esterification of lutein [132]. *Tritordeum* includes a high percentage of endosperm lutein esters and low quantities of durum wheat [124] or no lutein esters at all [133]. As a result, it was proposed that the high fraction of lutein esters in tritordeum has been generated by genes from *H. chilense* genome (Hch). Later, it appeared that XAT-7Hch was a prime match for lutein esterification in *H. chilense* and *Tritordeum*. Thus, the transfer of XAT-7Hch to wheat may be beneficial for increasing lutein esters in bio-fortification programs.

Moreover, colloidal delivery systems based on oil, water, and surfactants in the form of micro-emulsions, nanoemulsions, and standard emulsions, are essential for chemical preservation and delivery in food items and drinks. However, environmental friendly and practical techniques of esterification preparation are needed badly. Table 4 shows the stabilizing capacity of different matrices of lutein and zeaxanthin esters. In this respect, Shangguan et al. [134] described the synthesis of lutein esters utilizing a new covalently immobilized lipase B from *Candida antarctica* onto functionalized graphitic carbon nitride nanosheets. Gombač et al. [135] used viable and accessible procedures to create stable firm encapsulates of lutein and its esters. Lecithin emulsions exhibited higher encapsulation efficiencies, up to 99.3% for formulations having free lutein and up to 91.4% for formulations containing lutein esters. Due to the tiny droplet size, resistance to creaminess, and physical stability of lecithin emulsions, 17% and 80% of lutein and its esters were maintained within the formulations after 250 days.

The majority of commonly consumed fruit and vegetables contain far more concentration of lutein. However, high amounts of zeaxanthin (Z) and specifically zeaxanthin esters have also been reported in several fruits and vegetables [136]. For example, wolfberry, a fruit used to revamp eyesight in Chinese traditional medicine, has zea dipalmitate concentrations that can be 1 g/kg of the dry weight [137]. Moreover, chinese lanterns (*Physalis alkekengi*), orange pepper (*Capsicum annum*), and sea buckthorn (*Hippophae rhamnoides*) have been identified as reservoirs of zeaxanthin [136]. Breithaupt et al. [138] have reported a postprandial increase in plasma zeaxanthin when the subjects were fed with zeaxanthin dipalmitate meal than the meal having an equivalent amount of free zeaxanthin. Like lutein, Lam and But [139] have proposed that esterified zeaxanthin takes part in the durability of wolfberry’s therapeutic action. As a result of a recent investigation with rodents, esterified zeaxanthin is more bioavailable than the free form [140]. Humphries and Khachik [30] have indicated the total amount of all-trans- and multiple cis isomers of zeaxanthin in fruits after saponification of their extracts. However, there is relatively little information regarding the indigenous zeaxanthin ester pattern. Primary research has been done by Wingerath et al. [141] who studied MALDI mass spectrometry on zeaxanthin esters in tangerine juice. In addition to the esters of β-cryptoxanthin, zeaxanthin mono palmitate and five zeaxanthin diester(s) have been identified. In tangerine juice, however, the total concentration of zeaxanthin was relatively modest (1.3 mg/100g).

**Table 4 ijms-22-10910-t004:** Stabilization potential of lutein and zeaxanthin esters with different matrices.

Ester	Matrix	Results	Application	References
**Lutein**				
**ω-3 polyunsaturated fatty acid esters of lutein**	Fish oil	Better anti-oxidant ability and augmenting shelf-life	Oral supplements and nutraceuticals	[142]
**Lutein esters**	Nanoparticles	Improve ocular delivery efficiency	Oral drug formulation and eye-targeted drug delivery system	[143]
	Microcapsules	Increased half-life of lutein	For feed industry	[144]
	Polyvinylpyrrolidone	Improved shelf life and solubility	Antioxidant	[145]
	Polyoxyethylene sorbitan monooleate capsules	Improved stability	Ocular delivery system	[135]
**Zeaxanthin**				
**Zeaxanthin dipalmitate**	Sea Buckthorn oil and water emulsion	Increase bioaccesibility	Functional foods and nutraceutics	[146]
**Zeaxanthin ester**	Glycyrrhizic acid, arabinogalactan	Solubility enhancement	Food Industry	[147]

## 5. Thermal Degradation Kinetics and Cold Adaptability of Lutein and Zeaxanthin

The thermal degradation of lutein, β-carotene and β -cryptoxanthin in virgin olive oils is best illustrated by first-order kinetic process (VOO). According to Ruiz et al. [148], thermal stability varies across carotenoids, with lutein having higher thermal stability than β-carotene and β -cryptoxanthin with significant changes in geometric configuration. The isokinetic study examined kinetic and thermodynamic parameters (TP) in three Virgin olive oil (VOO) matrices with varying pigment concentration (high, medium, and low), and discovered that the oily medium had no effect on the reaction processes.

Pigment protein complexes (PPCs) of photosynthetic plants (PSA) protect injuries caused by disruption in the balance between light energy absorption and its use in photosynthesis. Leafy plants exposed to stress situations use numerous Chl and Car protection approaches when the light absorption surpasses the potential of utilizing the light reaction products. During the seasonal change, these mechanisms are coupled with substantial fluctuations in the qualitative and quantitative composition of photosynthetic pigments [149]. Recently, Sofronova et al. [150] have examined the periodic fluctuations in the ratio between photosynthetic pigments of one-year-old needles of Scotch pine (*Pinus sylvestris* L.). In July, when light and temperature were suitable, maximum accumulation of chlorophylls took place in developing young needles. The needles were noteworthy in this period because of their relatively high β-carotene and neoxanthin levels and also for their diminished lutein and violaxanthin-cycle pigments (VXC). In autumn hardening, the chlorophyll content decreased twice, but the total carotenoid concentration remained the same. In the start of hardening at lower and above-zero temperatures, there is a decrease in the β-carotene contents in the needles and the pigment protein complex (PPC) enrich with lutein.

This analysis predicts that when the temperature falls seasonally in the initial phases of hardening, the decline in chlorophyll content promote a decrease in the amount of absorbed radiant energy. Actually, the activation of lutein and neoxanthin in PPC antenna is followed by a gradual drop in the capacity of plants to satiate singlet energy for the excitation of chlorophyll. Due to the back reaction suppression of epoxidation at near-null temperatures, the accumulation of zeaxanthin allows indispensable prerequisites for turning on the mechanisms that allow absorbed light energy dissipation and do not depend upon the transmembrane proton gradient of thylakoids. Zeaxanthin can also serve as an antioxidant in the PPC and thylakoid membranes during the lipid phase [150]. Such results indicate that the observed reactions are adaptive, and individual pigments play a specific role in photosynthetic machinery’s structural and functional reorganization during the development of frost resistance in the needles.

## 6. Medicinal Applications of Lutein and Zeaxanthin

Carotenoids provide a wide range of nutritional and health benefits due to their multifarious biological impacts on humans, which include antioxidative, immunomodulatory, and anti-inflammatory properties [151]. Following are some of the effective implications of lutein and zeaxanthin in human beings.

### 6.1. Role of Lutein and Zeaxanthin in Treatment of Cognitive Impairment

Lutein and zeaxanthin account for 66–77% of the total carotenoid population in the brain, making them the main players in eye and brain health when compared to other members of the carotenoid family [151]. Lutein and its isomer zeaxanthin have a neurological connection between the macula and the brain, implying a role in cognition [152]. Lutein and zeaxanthin, as MP, are possible biomarkers of brain xanthophyll concentrations [153], and therefore provide information on the concentrations of lutein in the brain, making them a critical biomarker [154]. This includes the relationship with several measures of temporal processing speed [155], which is a necessary component of sensory and cognitive functions such as language, executive function, learning, and memory. A double-blinded, placebo-controlled trial on the cognitive implications of women who were given lutein dose (12 mg/d), docosahexaenoic acid dose (800 mg/d), or their combination for the period of 4 months revealed the significant improvement in verbal fluency scores in all 3 groups with boost in memory scores and learning rate [156]. Lutein and zeaxanthin have also been found to improve gap junctional communication [157], which is necessary for light processing and the development of neural circuitry in the visual system in the retina. These are also related with an increase in visual processing speed and a reduction in scotopic noise [158].

### 6.2. Role of Lutein and Zeaxanthin in Diabetes

Lutein may be useful in reversing the diabetic renal injury and inflammation through its anti-apoptotic and free radical scavenging properties. Diabetic nephropathy is a complex condition that results in the formation of free radicals, which act as a potent stimulant for pro-inflammatory factors. Lutein in the diet results in a reduction in serum and urine kidney function tests (urea and creatinine), indicating that diabetes-associated renal impairment is ameliorated. Different dose levels significantly reduce kidney inflammatory responses by decreasing TNF-, IL-1, and IL-6 (inflammatory mediators) levels while increasing IL-10 levels (anti inflammtory cytokine). Its supplementation at a greater dose, particularly the higher dose, ameliorate glomerular and tubular diabetes-induced damage and inflammatory infiltrates of renal leucocytes, therefore reducing oxidative and nitrosative stressors in renal tissue [159]. Patients with diabetic retinopathy (DR) group were given 6 mg/d of lutein with 0.5 mg/d of zeaxanthin for a period of 3 months with 0.0714 ± 0.0357 µg/mL of lutein and 0.0119 ± 0.0072 µg/mL of zeaxanthin in the control. Consequently, serum L/Z ratio experienced increase from 0.2816 ± 0.0731 µg/mL to 0.5409 ± 0.1807 µg/mL after L/Z supplementation as compared to normal individuals. Moreover, a decrease in average foveal thickness was also observed in 83% of patients. Thus, improving diabetic macular edema, visual acuity (VA) and contrast sensitivity (CS) in DR [160].

### 6.3. Role of Lutein and Zeaxanthin in Cancer

Lutein and α-carotene neutralize peroxy radicals through their antioxidative properties [161]. Carotenoids are combined to produce a stouter antioxidant defense as compared to their individual effect. Lutein or lycopene had the sturdiest effect, when combined synergistically [162]. Lutein depict its anti-carcinogenic activity by interacting with the mutagens including 1-nitro pyrene and aflatoxin B1 [163]. Moreover, the aforementioned carotenoid tends to activate specific genes that are involved in T-cell transformations after the spasm of various mitogens, cytokines, and antigens. Slattery et al. [164] detected a negative correlation between dietary lutein consumption and colon cancer in males and females.

Plasma lutein levels in humans are inversely related to cytochrome CYP1A2 activity, an enzymes involved in the metabolic instigation of putative human carcinogens in liver [165]. Lutein has been shown to have chemopreventive action in animal models of colon and breast cancer [166]. Zeaxanthin, lutein, and other carotenoids have been demonstrated to inhinit, dose-dependently. Haegele et al. [167] reported that plasma lutein levels in females is inversely related to oxidative DNA damage (8-hydroxy-2 deoxyguanosine (8-OHdG) in peripheral lymphocytes and lipid peroxidation.

#### 6.3.1. Treatment of Ovarian Cancer:

Increased uptake of zeaxanthin and lutein has appeared to be associated with 79% reduction in ovarian cancer. A number of epidemiologic investigations [168] has stated that zeaxanthin with lutein supplementation reduced ovarian cancer risk by 40–55 percent. In the Korean population, plasma levels of micronutrients such as ß-carotene, lycopene, zeaxanthin, lutein, retinol, and tocopherol have a role in lowering the risk of ovarian cancer [169]

#### 6.3.2. Treatment of Colorectal Cancer:

Ras signaling pathway appears to play a role in lutein-induced chemoprevention. Lutein has been known to lower the levels of many proteins involved in cell proliferation, notably K-ras and activated Akt (pPKB), in tumors of animals. Wang et al. [170] demonstrated that Inhibiting PI3K/Akt elicits cell cycle arrest and differentiation in a number of cell types, including the human colon cancer cell lines HT29 and Caco2. On the other hand, mice fed with lutein integrated into their diets and administered with Dimethyl hydrazine (DMH) demonstrated a substantial reduction of aberrant crypt foci (ACF) formation in the colon, which was associated with a decrease in cellular proliferation in terms of 5-bromo-2’-deoxyuridine. As compared to DMH-treated samples, lutein administration before and during DMH treatment reduced the concentration of β-catenin in colonic samples, hyperplasia, and adenocarcinomas, by acting as an efficient blocking agent.

#### 6.3.3. Treatment of Breast cancer

According to Rock et al. [171] an increased lutein concentration in plasma is directly linked with an increased chance of estrogen receptor positivity in females diagnosed with breast cancer. Lutein substantially reduces the incidence of breast cancer metastasis by causing cell-cycle arrest and caspase-independent cell death, which is quantitatively comparable to the effects caused by chemotherapeutic taxanes and docetaxel. Therefore, it can be stated that exposure of lutein plus taxanes decreases breast cancer cell growth [172]. Lutein results in ROS production in triple-negative breast cancer (TNBC) cells and induce growth-inhibition by its attenuated radical oxygen scavenger, N-acetyl cysteine, thus depicting an ROS mediated growth inhibition on TNBC cells. Moreover, lutein treatment activates the p53 signaling pathway and increases HSP60 levels, both of which contributes to the growth inhibition in TNBC cells [173]. Low level of dietary lutein i-e 0.002 and 0.02% fed to mice inhibited the incidence of mammary tumor, its growth as well as latency [174].

### 6.4. Role of Lutein and Zeaxanthin in Hepatic Disorders

Alcoholic liver diseases (ALD) are defined as hepatic injuries induced by excessive alcohol consumption (>20 g ethanol per day) and ranged from steatosis, alcoholic fatty liver disease (AFLD), and alcoholic hepatitis, to alcoholic cirrhosis [175]. Since zeaxanthin dipalmitate (ZD) is the most significant constituent of wolfberry, it is involved in the treatment of AFLD. ZD targets the key membrane receptors P2 × 7 and adiponectin receptor 1 (adipoR1) with a dose range of 10mg/g in chronic AFLD rat models. P2 × 7 and adipoR1 signals therefore regulate the phosphatidylinositol 3-kinase-Akt and/or AMP-activated protein kinaseFoxO3a pathways by restoring the mitochondrial autophagy (mitophagy) that was previously suppressed by ethanol intoxication [176]. Furthermore, ZD lessens hepatic inflammation by blocking Nod-like receptor 3 inflammasome. ZD may also help to reduce and prevent tissue scarring. Oral ZD (25 mg/kg) prevents secondary fibrosis and lowers collagen levels (including 4-hydroxyproline) in the liver. ZD also restore glutathione S-transferase (GST) activity, which has been proven to bind and neutralize bilirubin ZE (25 mg/kg) also inhibited liver fibrosis [177]. However, the underlying mechanisms remain unknown.

### 6.5. Role of Lutein and Zeaxanthin in Pregnancy

Carotenoids, notably lutein and zeaxanthin, are essential for the development of vision and the neurological system. These carotenoids are required for the improvement of the retinal activity, energy metabolism, and brain electrical activity, along with a few other functions [178].

Due to the absence of carotenoids in the majority of infant formulae, the carotenoid status of neonates and toddlers is dependent on the mother’s nutritional condition and also on infant feeding method. Increased mother L/Z consumption during pregnancy was related with improved verbal intelligence and capacity to regulate behavior in mid-childhood, indicating a possible advantage during prenatal development [179]. Many pregnant women take dietary supplements which in turn increases the plasma carotenoid levels [180]. The Norwegian Mother and Child Cohort Study (MoBa) revealed that the women who were not taking supplements has a lower plasma levels of carotenoids, i.e., 1.0 ± 0.50 µmol/L as compared those who were taking dietary supplements [181].

### 6.6. Role of Lutein and Zeaxanthin in Treatment of COVID-19

COVID-19 severity is increased by oxidative stress and inflammation, particularly in the presence of chronic illnesses associated with the antioxidant system’s fragility. As it develops, it causes hypoxemia, breathing difficulties, and acute respiratory distress syndrome. Some of these severe instances have high levels of pro-inflammatory cytokines (IL-6, IL2, IL7, IL10, GCSF, IP10, MCP1, MIP1, and TNF) and end in a deadly consequence [182]. NOX2-derived ROS, in particular, have been attributed to clotting and platelet activation, which either increases thrombin production and platelet aggregation or impairs arterial dilatation and endothelial dysfunction [183]. Carotenoids in diet are therefore known for quenching of ROS, like singlet oxygen and lipid peroxides, within the cell membrane’s lipid bilayer. Some clear evidences indicated that high amounts of α- and β- carotene, lutein/zeaxanthin, and total carotenoids are strongly linked with reduced incidence of oxidative stress, inflammation, and vice versa [184].

## 7. Lutein and Zeaxanthin Mediated Biological Pathways

### 7.1. Antioxidative Pathway in Macula

Since, the majority of lutein and zeaxanthin are present in human macula, here GSTP1 (glutathione S-transferase P1) and StARD3 (also known as steroidogenic acute regulatory domain 3) have been recognized as zeaxanthin and lutein binding proteins in human macula that allow the preferential accumulation of these carotenoids in humans. These proteins carry certain amino acid residues and loops that are matter of interest in binding with carotenoids (shown in Figure 5 for StARD3). However, the age-related decrease in the relationship between lutein and StARD3 could lead towards the primary cause of blindness in currently aging societies, which is age-related macular degeneration (AMD). This condition has two subtypes: wet AMD, which produces persistent vision damage despite therapeutic intrusions, and dry AMD, which has no particular treatment and causes progressive vision damage. Recently, Age-related Eye Disease (AREDS) and AREDS2 studies have undergone several clinical studies for preventive therapies. In this regard, AREDS2 has identified lutein as the carotenoid supplement to improve the positive impacts of the multi-vitamins and zinc, which appeared to be efficacious for the unique AREDS [185].

Exposure of retinal pigment epithelium (RPE) to disproportionate light energy has already been recounted to interrupt tight junctions. Light exposure also causes monocyte chemotactic protein-1 (MCP-1) synthesis, crucial for AMD due to its macrophage recruitment effects in the RPE and the choroid [186]. In this regard, Kamoshita et al. [187] have stated that lutein treatment encourages repair of photo-induced tight junction disruption in mice. Lutein lowers reactive oxygen species, enhances endogenous antioxidant dismutase (SOD) activity, and allows a continuous increase in *sod1* and *sod2* mRNA in the photo stressed RPE and neighboring choroid tissue. In addition, lutein reduces macrophage-related induction of RPE-choroid tissue markers (f4/80 and mcp-1). These results revealed that lutein supports tight junction repair and suppress inflammation in photo-stressed mice, lowering local oxidative stress through direct scavenging and, more preferably, through endogenous antioxidant enzymes inductions.

In addition to solar stressed conditions, Nuclear factor erythroid 2-related factor 2 (Nrf 2) signals get impaired in the aging RPE and thus become more prone to oxidative stress [188]. Transcription factor Nrf 2 regulates genes encode anti-oxidative and Phase II enzymes that are essential in cellular redox status maintenance and xenobiotic detoxification [189]. Under unconstrained conditions, Nrf 2 transcription activity is suppressed by the Kelch-like ECH associated protein 1(Keap1), resulting in the Nrf 2 proteasomal degradation. When the Nrf 2 pathway is activated, Nrf 2 translocates to the nucleus, creates a small musculoaponeurotic (Maf) fibrosarcoma heterodimer, and activates ARE-controlled genes [190]. The activation of Nrf 2 in human retinal pigment epithelial cell lines (ARPE-19 cells) is previously attributed to carotenoids like zeaxanthin and astaxanthin [191]. However, the role of lutein remains largely elusive in this activation. Recently, Frede et al. [192] have examined the effect of lutein on activation of nuclear factor erythroid 2 in ARPE-19 by lutein loaded Tween40 micelles. Since micelles were nontoxic with a concentration of ≥0.04 percent Tween40 in APRE-19 cells, thus caused cellular lutein accumulation and were identified as a suitable delivery system. As a result, lutein considerably boosts Nrf 2 translocation in the nucleus by 1.5 ± 0.4 times compared to unloaded Lutein micelles. This work together shows that lutein does not only work as a direct antioxidant, but also active Nrf 2 in the cells of ARPE-19.

Similar to macular degeneration (AMD), proliferative retinopathy (DR) is also associated with an increased level of vascular endothelial growth factor (VEGF) in the eye, which ultimately results in increased development of new blood vessels through a phenomenon of neovascularization. For proliferative DR, pan-retinal photocoagulation therapy is considered to be the gold standard for treatment to preclude severe vision loss. However, the therapy has a damaging effect and leads to the loss of possible photoreceptors and evening vision [193]. For this purpose, Keegan et al. [194] have used an in vitro cell model of the retinal microvascular endothelium to determine the effect of certain drugs on neovascularization processes. Their findings illustrate that lutein and zeaxanthin diminish tube formation that VEGF triggers. These substances have the potential to block the neovascularization of VEGF in 5:1 ratio considerably. These carotenoids, individually or in combination, can reduce oxidative stress caused by VEGF induction and increase the activity of NADPH oxidase, Nox4 [195]. Furthermore, it appeared that Nox4 inhibitors could reduce the protective effect of L and Z. These findings together demonstrate the protective role of L and Z in reducing VEGF-mediated neovascularization via a Nox4-dependent pathway.

#### Role of Human Macular Carotenoids in Reducing Alu RNA during Dry AMD

The dysfunction of the miRNA-processing enzyme, DICER1, and the accumulation of Alu RNA are associated with the pathogenesis of dry AMD. This finding is unique for dry AMD, since DICER1 levels in the RPE of human donor eyes and other retinal disorders like vitelliform macular dystrophy, retinitis pigmentosa, and retinal detachment, was not reduced [196]. There exists mounting evidence regarding the role of cellular oxidative processes in disease progression. Hydrogen peroxide (H_2_O_2_) is regarded as one of the powerful oxidants in ocular tissues [197]. In vitro studies have demonstrated that H_2_O_2_ exposure causes apoptosis and decreases cell viability in RPE [198]. Notably, H_2_O_2_ has been found to downregulate DICER1 mRNA in human RPE cells. Chong et al. [199] showed that lutein, but not zeaxanthin, had the potential to diminish the cytotoxic impact of H_2_O_2_ and the accumulation of Alu RNA in H_2_O_2_ challenged cells. However, before H_2_O_2_ exposure, when L was used alone or in conjunction with Z pre-treatment, it dramatically increased ARPE-19 cell survival and decreased Alu RNA levels in contrast to the negative control. In summary, these findings reveal a previously unknown role of H_2_O_2_ in the etiology of AMD and establish a foundation for further investigation of LUT and ZEA for preventing Alu RNA toxicity in RPE.

### 7.2. Zeaxanthin Mediated Apoptosis via ROS-Regulated Signaling Pathway in Human Gastric Cancer Cells

Gastric cancer is a significant public health issue and is categorized among the frequently occurring malignancies of the digestive system. Cancerous cells’ survival, differentiation, and migration are typically mediated by distinct intracellular signaling pathways, like the MAPK and AKT pathways. Members of the mitogen-activated protein kinase (MAPK) family, such as extracellular signal-regulated protein kinase (ERK), c-Jun N-terminal protein kinase (JNK), and p38 MAPK, are required for cell survival [200]. AKT is a serine/threonine protein kinase that may phosphorylate various proteins and is required for cell endurance. Research has demonstrated that the AKT and MAPK signaling pathways block the STAT3 signaling pathway [201].

Additionally, mitochondria create reactive oxygen species (ROS), i.e., superoxide anions, hydrogen peroxide, and hydroxyl radicals for signaling intermediates [202]. The formation of reactive oxygen species (ROS) is indeed one of the methods through which tumor cells are killed. Zeaxanthin signal transduction pathways play a critical role in generating reactive oxygen species (ROS) in the human gastric cancer cells [203]. Because control of intracellular ROS is critical for apoptosis, increased amounts of ROS inside the cell can trigger apoptosis via the MAPK pathway [204]. Previous research has established that although zeaxanthin can inhibit the development of reactive oxygen species (ROS) in normal cells, it can also promote ROS generation in melanoma [205]. In this regard, Sheng et al. [203] demonstrated that zeaxanthin induces apoptosis by lowering the mitochondrial membrane potential and increasing the expression of Cytochrome C, Bax, cleaved-caspase-3 (clecas-3), and cleaved-PARP (cle-PARP) while decreasing the expression of Bcl-2, pro-caspase-3 (pro-cas-3), and pro-PARP. Furthermore, zeaxanthin induced G2/M phase arrest by raising p21 and p27 levels and decreasing AKT, Cyclin A, Cyclin B1, and Cyclin-dependent kinase 1/2 (CDK1/2) levels. Nevertheless, both ROS scavenger N-acetylcysteine (NAC) and MAPK inhibitors prevented zeaxanthin-induced apoptosis, and MAPK regulated NF-B and STAT3 signaling pathway by decreasing their protein expression levels in the presence of zeaxanthin.

## 8. Industrial Applications of Lutein and Zeaxanthin

Despite having oncogenic, immunosuppressive, and non-eco-friendly properties, industrialization has accelerated the development and widespread usage of synthetic colorants in various fields of life. However, microbial pigments are gaining interest compared to synthetic colors because of their greater output, ease of downstream processing, and longer shelf life [206]. Although the recent market share of carotenoids is unknown, numerous market research surveys have indicated a definite upward tendency in the carotenoid industry. According to a BBC study, the global carotenoid industry-valued $1.5 billion in 2017 and is projected to reach $2.0 billion by 2022 [207]. Approximately 60% of the overall market share for carotenoids is comprised of β-carotene, astaxanthin, and lutein [208]. The following are the industrial uses of lutein and zeaxanthin:

### 8.1. Textile Industry

Among other industries, textile is the primary user of diverse dyestuffs. According to Lebeau et al. [209], the global dyestuff business demands a more significant proportion of natural pigments, which has resulted in the creation of natural colorants from plants and microorganisms. Natural colors derived from fungus have been demonstrated to be the best substitute of synthetic pigments due to their prompt development, ease of processing, and critical roles in transcription and intercellular communication. By extracting natural colorants from *Cassia tora* seeds, *Eucalyptus* sp. and *Grewia optiva* leaves in an aqueous media under various conditions, Dayal and Dobhal [210] have imparted their shades to silk, cotton, and jute textiles. Recently, it has been discovered that the blooms of *Tecoma stans* contain a fast-color yellow pigment called zeaxanthin. However, the main limitation of using plant-based dyes in today’s textile industry involves maintaining the color strength and fastness of fabrics dyed with natural colorants. In this regard, Adeel et al. [211] showed that 30 kGy gamma-ray treatment could increase the color intensity and fastness of lutein isolated from Marigold (*Tagetes erecta* L.) flowers. These findings indicate that gamma ray-induced lutein extraction from marigold flowers can work as a natural dye to generate yellowish-green shades.

### 8.2. Feed Industry

The natural pigmentation of poultry (particularly broilers) is mainly influenced by their diet’s composition, particularly by the carotenoid content. Lutein is currently commercialized globally as a feed supplement for poultry [212]. Commercially, marigold flower extract is added to poultry feed to enhance the pigmentation of birds (fat and skin) and egg yolks [213]. Additionally, partial cleavage (40–60%) of esterified pigments may occur within the broiler’s stomach [214]. Saponification is typically conducted during commercial preparations to increase the extract’s pigmentation value. Once ingested, free (unesterified) lutein is transported and stored as diesterified lutein in the liver or integumentary tissues [215]. The consumer frequently associates a brilliant yellow color in egg yolks, skins, and fatty tissues with excellent health and premium quality. Moreover, feeding lutein may increase the production performance of yellow catfish and the reproductive performance of quail [216]. Additionally, zeaxanthin is utilized in the food sector as a feed additive and colorant for birds, pigs, and fish. The pigment gives birds’ skin and egg yolks a yellowish color, while it is used for skin pigmentation in pigs and fish [217].

### 8.3. Cosmetics Industry

By successfully decreasing the quantity of oxidative stress in the skin, topical application of lutein unintentionally aids in moisture retention and anti-aging effects [218]. Marigold flower extract (*Tagetes erecta*) is primarily used to produce a brand of lutein called FloraGLO^®^ lutein, which is marketed by Kemin industries. FloraGLO^®^ lutein could be taken orally or applied topically. Ten milligrammes of FloraGLO^®^ lutein daily improves skin elasticity, hydration, and lipid content [219]. However, the chemical processes behind the positive effects of xanthophylls (lutein/zeaxanthin) on the skin remain unclear. But, Li et al. [121] have recently demonstrated that lutein and zeaxanthin modify gene expression by stimulating hyaluronan synthesis in a human keratinocyte model. As a result, these modifications provide a mechanistic basis for the clinical benefits of xanthophylls.

## 9. Future Perspective

The global nutraceutical market is projected to extensive growth due to an increased demand for healthy and organic food supplements. It has been noted that the amount of zeaxanthin and lutein in supplements should be balanced to successfully increase macular pigment optical density and effectivity of antioxidant pathways. The everlasting effects of L and Z status in reducing the menace for age-related and other maladies can be apprehended in long-term epidemiological studies. Numerous approaches are currently on their way, but the actual pooling of statistics from accessible cohorts can promote investigations to the degree where phenotypes and genotypes may alter L and Z levels in dietary supplements for optimal benefits over a wide range of intake levels.

Although various natural sources are available for lutein and zeaxanthin production, the need of an hour is to instigate industrial involvement in research and development activities for bacterial lutein and zeaxanthin production and other value-added bio-products, using a bio-refinery approach. For more than a century, it has been speculated that isoprenoid pathways are the most palpable targets for the laboratory evolution of xanthophylls. Most laboratory-evolved and engineered isoprenoid pathways have so far accumulated unique carotenoids as a component of complicated assortments. This may be adequate for the discovery of new carotenoids. Still, to provide valuable insights for a synthetic route to a specific compound, the pathway must be further engineered to become more precise. Usually, the natural carotenoid pathway tends to produce only a few end products that probably follow a similar course of “pruning.” Several possible approaches, including the engineering of carotenoid biosynthesis module that mainly include the choice and engineering of unique enzymes, localized expression of key enzymes, optimization of gene expression and enhancing the carotenoid stowage in producing cells, have been enlisted to transform nonspecific pathways into the ones that upsurge the metabolic flux in the direction of target production, i.e., lutein and zeaxanthin. However, still, an in silico metagenomics approach needs to be underpinned for this purpose. In this regard, omics analysis entails the valuable means to comprehend the whole cell activity and can further expand carotenoid production.

A comparative proteomic examination of excessive lutein and zeaxanthin producing strains [220] and transcriptomic analysis of a lycopene strain using fructose as carbon source [221] acknowledge synchronized pathways and genes that can be additionally improved for carotenoid synthesis. Moreover, lutein and zeaxanthin synthesis pathway may engineer possibly after the accumulation of neutral mutations (in the context of the desired pathway). In this regard, a number of genetically engineered microbial strains are widely used in industrial processes for enhanced lutein and zeaxanthin production. But pronounced attempts are required for microbial carotenoid over production since, the genetic stability of microbial strains limits their applications in industry. Only a few bioprocesses that emphasis on chromosomal change address such limitations but most of them still use plasmids for gene regulation that are less stable and require antibiotics which tolerate higher metabolic burden [222]. Therefore, a number of policies have been employed to unravel these problems in which novel marker-free genetic integration tools based on the CRISPR-Cas system are the matter of interest. Although the production of zeaxanthin is enhanced through this aforementioned tool but lutein production needs to be enhanced through this technique too.

Moreover, C35, C45, and C50 carotenoid pathways discovered in laboratories allow sophisticated experimental efficiencies for validating the production of lutein and zeaxanthin. In this regard, investigations directed at evolving pathway specificity in the laboratory would permit researchers to track the effect of each genetic change on the esterification and synthesis of lutein and zeaxanthin. More research focused on the use of agro-industrial wastes, high lutein, and zeaxanthin producing organisms through metabolic engineering strategies, bioreactor design and control strategies for microbial carotenoids production is necessary to cover the worldwide demand of lutein and zeaxanthin for poultry, cosmetic, food, feed, beverages and others industries.

## 10. Conclusions

Lutein and zeaxanthin are the typical xanthophylls that improve food quality due to their high abundance in five kingdoms of life with major availability in Kingdom *Plantae* and *Protista*. Although marigold flower and certain eubacterial strains meet the global demand for lutein and zeaxanthin to some extent, there is still a huge opportunity to contribute to the global demand for natural lutein and zeaxanthin. For this purpose, a number genetic engineering techniques including random mutagenesis and gene assembly and CRISPR/Cas9 based approaches have been employed to enhance the productivity of lutein and zeaxanthin but a metagenomics approach needs to be implied for the pronounced applications. Numerous investigations, both preclinical and clinical, have indicated that lutein and zeaxanthin can slow the course of retroviral, malignant, and ocular disorders, mostly by quenching free radicals and protecting them from oxidative damage. According to study findings, lutein and zeaxanthin can effectively mitigate oxidative stress in vivo and protect the eye. Additionally, due to their beneficial bioactive characteristics, the concept of integrating lutein and zeaxanthin into meals, nutraceuticals, or cosmetic is gaining attraction.

## Figures and Tables

**Figure 1 ijms-22-10910-f001:**
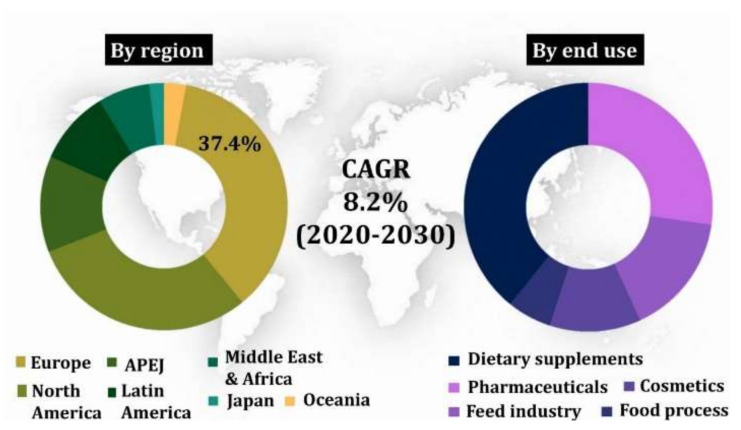
Global Zeaxanthin Market share by region and end use (2020–2030) with compound annual Growth rate of 8.2%. Source: (Transparency Market Research, 2020) [14].

**Figure 2 ijms-22-10910-f002:**
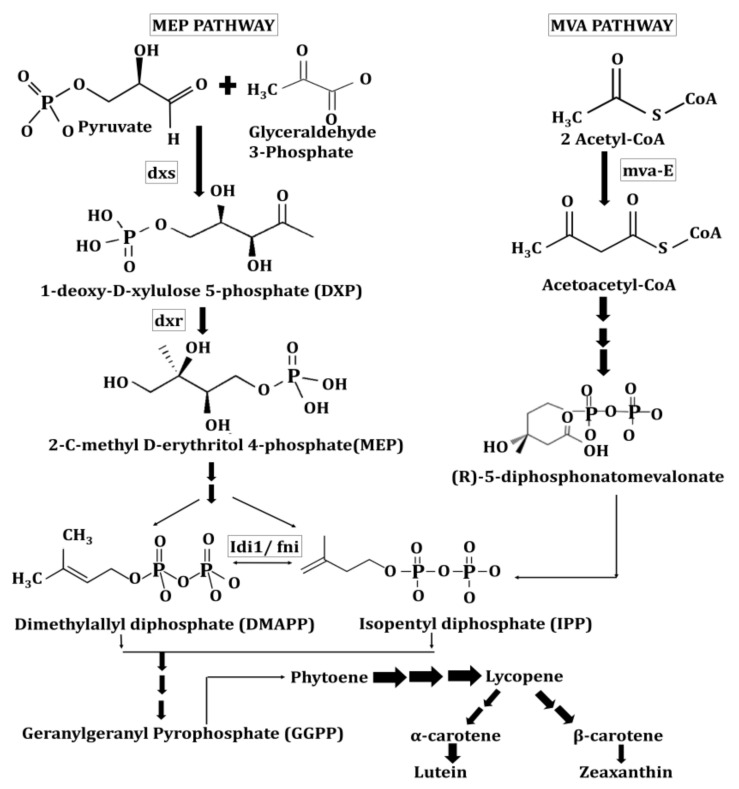
A simplified schematic representation of Biosynthesis of Lutein and Zeaxanthin via both Methylerythritol4-phosphate (MEP) and mevalonate (MVA) pathway. DMAPP and IPP, precursors for α-, β-carotene are synthesized via both of these pathway. Enzymes involved in the process include: dxs: deoxyxylulose 5-phosphate synthase; dxr, deoxyxylulose 5-phosphate reductoisomerase; idi1: Isopentyl-diphosphate-isomerase 1; mva-E: Acetyl Co-A acetyl transferase.

**Figure 3 ijms-22-10910-f003:**
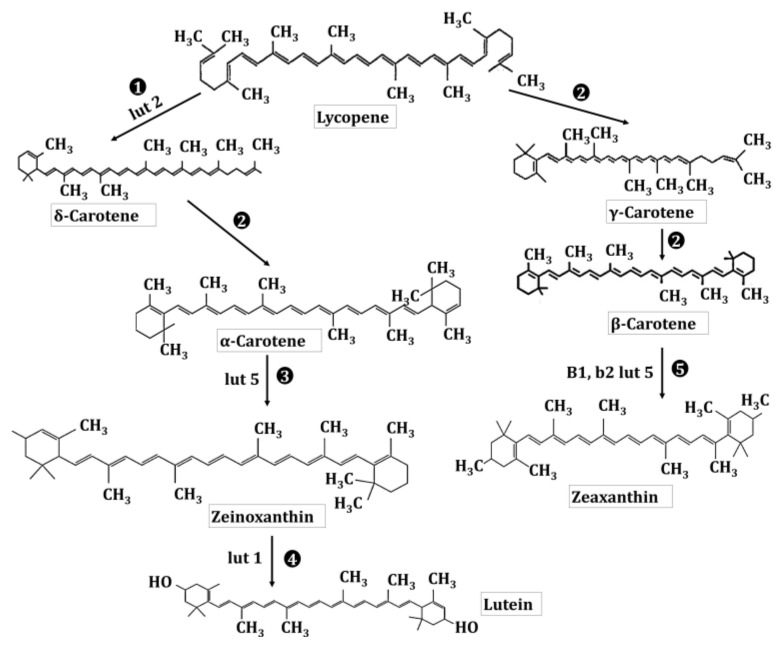
Pathway showing the most preferred routes of lutein biosynthesis in *Arabidopsis.* Enzymatic reactions are indicated by numbers; **1**: ε-cyclization by ation, **3**: β-hydroxylation of carotenoids, **4**: ε-ring hydroxylation, **5:** β- ring hydroxylation of β, β-carotenoids by CYP97A3 (*lut* 5 locus). Modified from Kim et al. [26].

**Figure 4 ijms-22-10910-f004:**
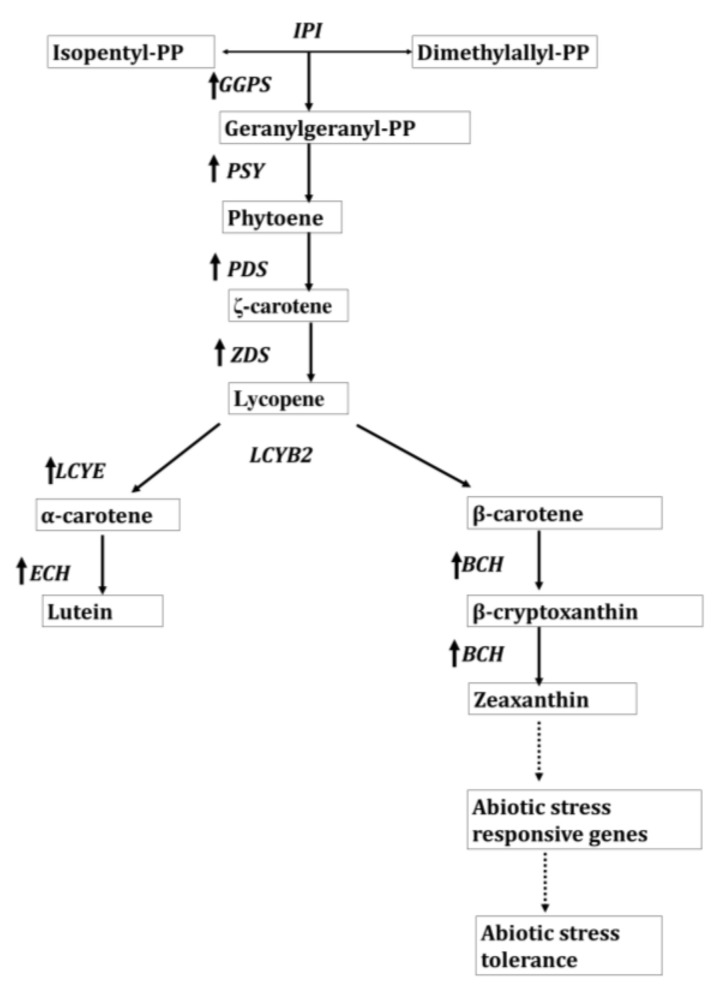
Diagram shows a proposed model for boosting carotenoid biosynthesis and antibiotic stress responses in the transgenic sweet potato plants. The upregulation of enzymes i.e., IPI (isopentyl-diphosphate delta isomerase), GGPS (Geranylgeranyl pyrophosphate synthase), PSY (Phytoene synthase), PDS (Phytoene desaturase), ZDS (Zeta-carotene desaturase), LCYE (Lycopene epsilon cyclase), ECH (Enoyl-CoA hydratase), BCH (Beta carotene hydroxylase. Modified from Kang et al. [119].

**Figure 5 ijms-22-10910-f005:**
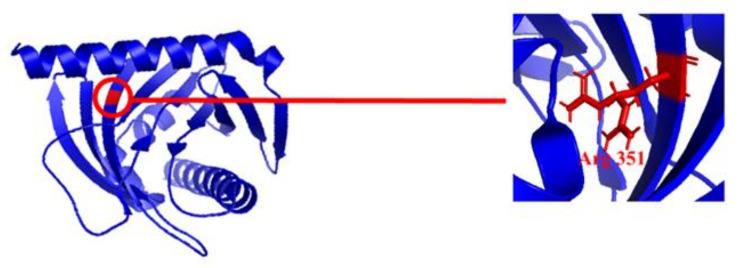
Computational Structure of StARD3 (PDB IB: 5I9J). Along with Ω loop, Arg 351 (shown in red sticks) is appeared to be an important amino acid residue involved in binding with lutein and allow the retention of lutein in human macula.

**Table 2 ijms-22-10910-t002:** Production of lutein and zeaxanthin by micro-algal species.

Microalgae	Culture Conditions	Biomass	Lutein Yield	Zeaxanthin Yield	Other Carotenoids	References
**Division Chlorophyta**						
* **Chlorella pyrenoidosa** *	Fresh water cultures	N.A	125034.4 µg/g	2170.3µg/g	cis isomers of β-carotene, all trans- β -carotene, cisisomers of α-carotene, β –cryptoxanthin, neoxanthin and its cis isomers, neochrome, auroxanthin, violaxanthin and its cis isomers	[63]
* **Chlorella sorokiniana** *	Fresh water cultures	1.98 g/L/d	6490 µg/g)	N.A	Astaxanthin	[64]
* **Chlorella salina** *	Fresh water cultures	665.89 mg	9.73 mg/L/day	N.A	N.A	[65]
* **Chlorella zofingiensis** *	Fresh water cultures	7 g/L	4000 µg/g	7000 ± 820 µg/g	monoester of astaxanthin and canthaxanthin	[66,67]
* **Chlorella vulgaris** *	Fresh water cultures	N.A	3360 µg/g	N.A	Astaxanthin 12.5% TC violaxanthin	[68]
***Dunaliella salina* zea1 (mutant)**	Fresh water	N.A	N.A	200 µg/g	Astaxanthin and violaxanthin	[69]
* **Chlorella protothecoides** *	Fresh water	19.6 g/L	68.42 and 83.81 mg/L	N.A	N.A	[70]
**Division Rhodophyta**						
***Porphyra acanthophora* var. *brasiliensis***	Saline condition	N.A	4.16–30.71 µg/g	4.43–36.31 µg/g	N.A	[71]
***Pyropia yezoensis*:**	Marine culture		3460 570 µg/g	2.36–17.58 µg/g	α-carotene, α-cryptoxanthin (β,ε-caroten-3′-ol), and zeinoxanthin (β,ε-caroten-3-ol)	[72]
***Porphyra* sp.**	Marine culture	N.A	430–1117 µg/g	N.A	N.A	[73]
* **Corallina elongata** *	Marine culture	N.A	1.3 ± 0.6%	4.7 ± 0.1%	Anteraxanthin	[74]
* **Rhodymenia corallina.** *	Marine Culture	N.A	3.4 ± 0.3 µg/g	N.A	Trans-β-carotene, Cis-β-carotene,	[75]
**Division Euglenophyta**						
* **Euglena gracilis** *	Fresh water	0.5 g/L	700 mg/100g	N.A	Neoxanthin and β-carotene.	[76]

**Table 3 ijms-22-10910-t003:** Production of zeaxanthin in domain eubacteria.

Family Name	Microorganism	Major Carotenoid Content	Other Carotenoids	General Applications	References
**Flavobacteriaceae**	*Flavobacterium* sp. P8 strain	205 μg/g	β-cryptoxanthin and β-carotene	Prevention of macular degeneration	[87]
	*Flavobacterium multivorum*	(0.05 μg/mL/h)	β-cryptoxanthin and β-carotene	Antioxidant	[88]
		10.65 ± 0.63 μg/mL	β-cryptoxanthin and β-carotene	For commercialization of zeaxanthin based products	[89]
	*Flavobacterium* sp.	1 mg/L	β-cryptoxanthin and β-carotene	Natural food colorant for fish and poultry	[90]
	*Mesoflavibacter zeaxanthinifaciens*	0.91 mg/g	Yellow carotenoid pigments	For use as a natural colorant in feed industry	[91]
	*Mesoflavibacter aestuarii* sp.	1200 mg/100 mg	N.A	Antioxidant	[92]
	*Aquibacter zeaxanthinifaciens*	N.A	Some unidentified carotenoids	Antioxidant	[93]
**Enterobacteriaceae**	*E. coli* (metabolic engineered strain)	1.6 mg/g	β-carotene	Antioxidant	[94]
	*E. coli*	11.95 mg/g	β-cryptoxanthin and β-carotene	Antioxidant	[95]
**Sphingobacteriaceae**	*Nubsella zeaxanthinifaciens*	0.8 mg/g	N.A	Natural colorant	[96]

## Data Availability

Not applicable.

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
