# Peer review of "Biochemical and Immunological implications of Lutein and Zeaxanthin"

_ijms, 2021, doi:10.3390/ijms222010910_

Round 1

Reviewer 1 Report

In the manuscript “Biochemical and Immunological implications of Lutein and Zeaxanthin”, the authors propose to discuss the role of lutein and zeaxanthin in some biochemical and immunological mechanisms. The paper could be of interest but there are too many drawbacks that hampered the initial enthusiasm. The first part of the manuscript is too long and is written as a book chapter and not a review manuscript. The second part is too short and could be improved. There are some important issues to be solve. I have some specific comments that may be useful.

Specific comments:

  1. There are a few problems in figures. For instance, figure 2 is not with good quality. Please correct this.
  2. The first subchapters (up to 3.5) are very descriptive and written more like a book chapter than a revision of literature. The authors are encouraged to have an historical approach or build a hypothesis. Still, the authors must shorten these sections and merge some.
  3. The manuscript has too many chemistry and less molecular approach to the subject and thus, the authors should focus on molecular studies also. In addition, when discussing the studies, the authors must include species, concentrations, number of animals and other relevant information to discuss.
  4. The last part of the paper, which presents the effects on diseases (diabetes, cancer…) should be the focus of the paper. Please explain the major aspects of the disease and then explain the use of lutein and zeaxanthin. Explore the studies, explain and put it in perspective.
  5. Overall the paper is too long. The first part should be rewritten and shorten. The paper is also too vague and the rationale for the utility is not clear.

Author Response

Review comment: There are a few problems in figures. For instance, figure 2 is not with good quality. Please correct this.

Author’s Response: Figure 2 is removed from the manuscript as it appeared to be unnecessary in the review paper.

Review Comment: The first subchapters (up to 3.5) are very descriptive and written more like a book chapter than a revision of literature. The authors are encouraged to have an historical approach or build a hypothesis. Still, the authors must shorten these sections and merge some.

Author’s Response: These subchapters have been shortened to a certain point but in most of the reviews regarding lutein and zeaxanthin all the possible sources are not listed so this would be a plus point for the paper. Moreover, historical approaches have also been cited here in the manuscript.

Review Comment: The manuscript has too many chemistry and less molecular approach to the subject and thus, the authors should focus on molecular studies also. In addition, when discussing the studies, the authors must include species, concentrations, number of animals and other relevant information to discuss.

Author’s Response: Since, the article title involves the biochemical aspect of these xanthophylls therefore the manuscript carried biochemical approach in it. But due to the objection certain biochemical approaches have been removed from the article and molecular approaches are cited here. Also the concentration, species and number of subjects have been added in the section of medicinal implications and genetic engineering approaches.

Review Comment: The last part of the paper, which presents the effects on diseases (diabetes, cancer…) should be the focus of the paper. Please explain the major aspects of the disease and then explain the use of lutein and zeaxanthin. Explore the studies, explain and put it in perspective

Author’s Response: The use of lutein and zeaxanthin is explained in the manuscript and also added in perspective.

Review Comment: Overall the paper is too long. The first part should be rewritten and shorten. The paper is also too vague and the rationale for the utility is not clear

Author’s Response: All the authors have tried to shorten the paper but as it is carrying all the possible details regarding lutein and zeaxanthin, therefore, it is getting extensive.

Reviewer 2 Report

I have carefully read your manuscript entitled "Biochemical and Immunological implications of Lutein and Zeaxanthin". This is an extensive review on lipophilic isoprenoid pigments synthesized by plants, algae, bacteria and fungi, but not synthesized by humans - carotenoids.

The issue of the review is noticeable, suitable to the journal aims and scope. In my opinion, the manuscript needs only minor revision due to grammar, spelling and style mistakes. The text should have been checked by a native speaker (capital letters, missing commas, verbs past participle forms, verb forms for subject-verb agreement, wordy phrases, missing determiners etc.)

Keywords should include other words than a title.

Use subscripts for chemical formulas.

Do not use capital letters at the beginning of words in the middle of sentences.

In tables, consider arranging the results alphabetically.

Opt for a uniform number of digits in the numerical results.

Write shorter:

3.2. Prevalence in kingdom protista-algae 

3.3. Distribution in kingdom animalia

3.4. Prevalance in kingdom fungi

3.5. Biosynthesis in domain eubacteria etc.

8.1. Role in treatment of cognitive impairment

8.2. Role in diabetes etc.

Best regards,

Reviewer

Author Response

Reviewer comment: The issue of the review is noticeable, suitable to the journal aims and scope. In my opinion, the manuscript needs only minor revision due to grammar, spelling and style mistakes. The text should have been checked by a native speaker (capital letters, missing commas, verbs past participle forms, verb forms for subject-verb agreement, wordy phrases, missing determiners etc.).

Author’s response: All the grammar, spelling and style mistakes have been corrected.

Reviewer Comment : Keywords should include other words than a title.

Use subscripts for chemical formulas.

Do not use capital letters at the beginning of words in the middle of sentences.

In tables, consider arranging the results alphabetically.

Opt for a uniform number of digits in the numerical results.

Author’s Response: The comment is addressed properly.

Reviewer Comment : Write shorter:

3.2. Prevalence in kingdom protista-algae 

3.3. Distribution in kingdom animalia

3.4. Prevalance in kingdom fungi

3.5. Biosynthesis in domain eubacteria etc.

8.1. Role in treatment of cognitive impairment

Author’s Response: These headings have been shortened in the manuscript